# TTFSFormer: A TTFS-based Lossless Conversion of Spiking Transformer

Lusen Zhao [1]    Zihan Huang [1]    Jianhao Ding [1]    Zhaofei Yu[✉ 1]

## Abstract

ANN-to-SNN conversion has emerged as a key approach to train Spiking Neural Networks (SNNs), particularly for Transformer architectures, as it maps pre-trained ANN parameters to SNN equivalents without requiring retraining, thereby preserving ANN accuracy while eliminating training costs. Among various coding methods used in ANN-to-SNN conversion, time-to-first-spike (TTFS) coding, which allows each neuron to at most one spike, offers significantly lower energy consumption. However, while previous TTFS-based SNNs have achieved comparable performance with convolutional ANNs, the attention mechanism and nonlinear layers in Transformer architectures remains a challenge by existing SNNs with TTFS coding. This paper proposes a new neuron structure for TTFS coding that expands its representational range and enhances the capability to process nonlinear functions, along with detailed designs of nonlinear neurons for different layers in Transformer. Experimental results on different models demonstrate that our proposed method can achieve high accuracy with significantly lower energy consumption. To the best of our knowledge, this is the first work to focus on converting Transformer to SNN with TTFS coding. The source code of the proposed method is available at https://github.com/ForestOnTheLand/TTFSFormer.git.

## 1. Introduction

Spiking Neural Networks (SNNs), known as the third generation of neural networks (Maass, 1996), are promising due to their low computational cost and biological plausibility. Inspired by biological neurons, SNNs utilize binary spikes for communication between neurons instead of numerical values. Compared to traditional Artificial Neural Networks (ANNs), SNNs achieve higher energy efficiency due to the sparsity of event-driven signaling, particularly on neuromorphic hardware (Davies et al., 2018; DeBole et al., 2019; Pei et al., 2019).

The Transformer architecture (Vaswani et al., 2017) has been demonstrating its strong ability in a wide range of tasks with its special self-attention mechanism, from natural language processing to computer vision (Dosovitskiy et al., 2020). Recent efforts to integrate Transformers with SNNs seek to combine the information processing ability of Transformers with the energy efficiency of SNNs, achieving promising results in many tasks (Wang et al., 2023; Yao et al., 2023).

Currently, there are two primary approaches for training spiking transformers: direct training and ANN-to-SNN conversion. Direct training methods (Yao et al., 2023; Zhou et al., 2023b;a) rely on surrogate gradients (Neftci et al., 2019) to perform backpropagation on networks, which approximates the non-differentiable spike series with continuous functions (e.g., sigmoid or tanh) during gradient computation. However, direct training of large-scale SNNs from scratch (Yao et al., 2023; Zhou et al., 2023a) requires high computational resources, posing a major challenge in its scalability and efficiency. In contrast, ANN-to-SNN conversion (Wang et al., 2023; Jiang et al., 2024; Huang et al., 2024) maps pre-trained ANN parameters to SNN counterparts without requiring retraining, thereby preserving ANN accuracy while eliminating training costs. However, previous Transformer-to-SNN methods primarily rely on rate coding, where information is encoded through spike counts over time. A key limitation is that precise value representation often requires multiple spikes, which reduces energy efficiency.

Recent evidence from neuroscience suggests that biological neural systems encode information not only through firing rates but also via precise spike timing (Gütig & Sompolinsky, 2006; Montemurro et al., 2008; Park et al., 2019). Temporal coding methods leverage this observation by representing values through spike timing rather than spike counts. Notably, time-to-first-spike (TTFS) coding achieves high energy efficiency by encoding information in the latency of a single spike (Rueckauer & Liu, 2018). However, TTFS-

[1] Peking University, China. Correspondence to: Zhaofei Yu <yuzf12@pku.edu.cn>.

*Proceedings of the 42nd International Conference on Machine Learning*, Vancouver, Canada. PMLR 267, 2025. Copyright 2025 by the author(s).

based ANN-to-SNN approaches have so far been limited to Multi-Layer Perceptrons (MLPs) and Convolutional Neural Network (CNN). In this work, we propose the first TTFS coding framework for Transformer conversion, which simultaneously maintains high accuracy and reduces energy consumption through precise spike timing. Our main contributions are summarized as follows.

- We analyze the limitations of the representation ability of previous TTFS-based methods.

- We propose a generalized type of TTFS-based neuron, enabling the lossless conversion of all layers in the Transformer architecture into an SNN while significantly reducing energy consumption.

- We evaluate our method on various pre-trained Transformer models, including ViT and EVA, using the ImageNet-1K dataset. Experimental results demonstrate that our approach achieves performance comparable to ANN counterparts. Specifically, the accuracy loss of the converted ViT and EVA models, across different parameter sizes, remains below 0.1% with precise representation of TTFS spike timing.

- To the best of our knowledge, this is the first work to explore TTFS-based SNNs within the Transformer architecture.

## 2. Related Works

### 2.1. Transformer for Visual Tasks

With its specialized attention mechanism that captures contextual information, the Transformer (Vaswani et al., 2017) has become dominant across various fields of machine learning, including natural language processing and computer vision. By dividing an image into blocks and converting them into tokens, the Vision Transformer (ViT) (Dosovitskiy et al., 2020) successfully adapted the Transformer architecture for visual tasks, effectively processing spatial information. The EVA model (Fang et al., 2023; 2024) further introduced additional techniques, such as the gated linear unit and various activation functions, significantly enhancing performance.

### 2.2. Spikes Coding Methods in SNN

One of the key differences between SNNs and ANNs is that SNNs use spikes to transmit information between layers, whereas ANNs rely on floating-point numbers or integers. The representation method, which defines the mapping between the value and the pattern of emitted spikes, plays a crucial role in both encoding external inputs and decoding model predictions. This makes it a critical issue in SNN architectures. Considering the trade-off between performance and energy efficiency, previous studies have focused on

two primary representation methods in SNNs: rate-based encoding and temporal-based encoding.

Rate-based SNNs map the value to the frequency of spikes. Specifically, if a neuron emits $N$ spikes over $T$ time steps, it represents the value $\frac{N}{T} \in [0,1]$. Rate coding has been widely adopted in previous studies, demonstrating comparable performance to ANNs, including both directly trained SNNs and those converted from ANNs. However, rate-coding typically requires multiple time steps to efficiently approximate the value with spikes, which results in higher energy consumption.

Temporal-based SNNs, inspired by the temporal information observed in biological neural systems, map the value to the timing of spikes. These methods include time-to-first-spike (Thorpe et al., 2001), reverse coding (Zhang et al., 2019; Park et al., 2020), phase coding (Montemurro et al., 2008) and burst coding (Park et al., 2019). TTFS represents the value by the exact time of the single spike emitted, where a higher value results in an earlier spike. Reverse coding (Zhang et al., 2019; Park et al., 2020), in contrast, make the convention that the stronger the input stimulus is, the later the corresponding neuron fires a spike. Phase coding generates the post-synaptic potential (PSP) based on the phase of a periodic oscillatory function. Burst coding encodes information through a sequence of spikes with short inter-spike intervals. Temporal-based methods are typically more energy-efficient than rate-based methods, as they require fewer spikes to represent a value. However, despite the variety of temporal-based SNNs, previous studies have largely focused on simple network architectures, such as MLPs and CNNs.

### 2.3. ANN-to-SNN Conversion

To address the challenges posed by the non-differentiable spike generation function in SNN training, the ANN-to-SNN conversion method effectively leverages well-established techniques from ANN training to achieve high performance. Cao et al. (2015) initially proposed obtaining an SNN by training an ANN and then converting it. The resulting SNN conveys information through spike rates, commonly referred to as a rate-based SNN. Building on this, Diehl et al. (2015) reduced the accuracy gap between the source ANN and the converted SNN through weight normalization. Subsequent studies have further narrowed this gap by employing tailored activation functions or specialized adjustments to spiking neurons (Ding et al., 2021; Bu et al., 2022; Hao et al., 2023). It is important to note that these approaches are primarily based on deep CNNs.

With the growing significance of Transformer architectures across various tasks (Dosovitskiy et al., 2020; Fang et al., 2023), recent studies have also explored rate-based converted SNN Transformers. Wang et al. (2023) were the first

to apply the conversion method on Transformer architectures. Jiang et al. (2024) introduced the Spatio-Temporal Approximation (STA) method, which approximates nonlinear operations using multiple neurons, referred to as the universal group operator. STA bridges the gap between Transformers and SNNs but results in higher energy consumption and increased latency. Huang et al. (2024) further improved the performance by incorporating an expectation compensation module (ECM) and multi-threshold neurons, achieving lower accuracy loss with fewer time steps. However, multiplication operations still remain present in the activation layers.

While rate-based conversion has demonstrated high performance, some ANN-to-SNN conversions also leverage temporal-based encoding methods. Rueckauer et al. (2018) first proposed several temporal-based conversion methods. Zhang et al. (2019) explored the effective conversion of deeper ANNs into SNNs using TTFS coding, and Park et al. (2020) further improved the performance. Previous TTFS-based studies introduced conversion errors across layers until Stanojevic et al. (2023; 2024) demonstrated an exact mapping from ReLU network to SNN. Building on this, they further developed a TTFS-based training method. However, while TTFS-based SNNs can achieve performance comparable to their corresponding CNN models, no previous TTFS-based SNN has involved Transformer architecture, which can achieve superior task performance.

## 3. Preliminaries

In this section, we introduce key techniques from previous studies that implement TTFS encoding and enable TTFS-based ANN-to-SNN conversion. These techniques address the challenges of directly training SNNs while leveraging the advantages of TTFS encoding.

### 3.1. TTFS Encoding

Inspired by biological observations that precise spike timing is important for communications between neurons, TTFS-based SNNs can operate with at most one spike per time window per neuron by utilizing precise spike timing, achieving higher energy efficiency compared to rate-based SNNs (Rueckauer & Liu, 2018).

Recent studies have adopted a linear relationship between input values and spike times, effectively enabling the conversion of ReLU-based ANNs to SNNs while preserving accuracy and computational efficiency (Stanojevic et al., 2023; 2024). The linear relationship of TTFS between the spike time $t$ and the represented value $x$ is defined as

$$x = \frac{t_{\max} - t}{t_{\max} - t_{\min}}, \qquad (1)$$

where $[t_{\min}, t_{\max}]$ defines the available range of spike times,

ensuring that the corresponding value $x$ remains within $[0, 1]$.

### 3.2. TTFS-based Neurons

Inspired by biological neurons, TTFS-based neurons receive spikes from the previous layer and maintain a membrane potential $V$ that evolves over time. Once the potential reaches the threshold $\theta$, the neuron emits a spike. The temporal dynamics of the membrane potential in a TTFS-based neuron can be described by the following equation:

$$\tau \frac{\mathrm{d}}{\mathrm{d}t} V(t) = \sum_{i=1}^{N} w_i K(t - t_i) + C, \qquad (2)$$

where $t_i$ represents the presynaptic spike time of the $i$-th neuron in the previous layer, and $w_i$ is the corresponding synaptic weight. Each presynaptic spike induces a postsynaptic potential (PSP), modeled by the kernel function $K(\cdot)$. $C$ is a constant, and $\tau$ represents the time constant.

Various kernel functions have been proposed for constructing TTFS-based SNNs. Stanojevic et al. (2023) utilized the Heaviside function as the kernel function to establish an exact mapping from ReLU neurons to TTFS-based neurons:

$$K(t - t_i) = \mathcal{H}(t - t_i) = \mathbb{I}[t - t_i \geq 0], \qquad (3)$$

where $\mathbb{I}(\cdot)$ denotes the indicator function. The use of the Heaviside function enforces a linear relationship between presynaptic spike times and their corresponding postsynaptic spikes, making it challenging to construct complex nonlinear mappings.

Another recent work by Goltz et al. employed the current-based (CuBa) neuron model (Göltz et al., 2021), where the kernel function is defined as:

$$K(t - t_i) = \mathcal{H}(t - t_i) \exp(-\frac{t - t_i}{\tau}), \qquad (4)$$

where $\tau$ is a constant. This kernel function, also known as the alpha-PSP response function, facilitates the gradient computation with respect to spike times. Goltz et al. constructed a TTFS-based SNN using the CuBa neuron model and directly trained the network. However, the SNNs were only evaluated on small datasets, and their scalability remains unverified.

### 3.3. TTFS-based ANN-to-SNN Conversion Method

The TTFS-based ANN-to-SNN conversion method requires precise spike timing to represent the layer-by-layer activations of deep ANNs. Currently, an exact conversion approach involves dividing the activation process of spiking neurons into two stages: the **receiving stage** and the **emitting stage**. During the receiving stage, the neuron receives

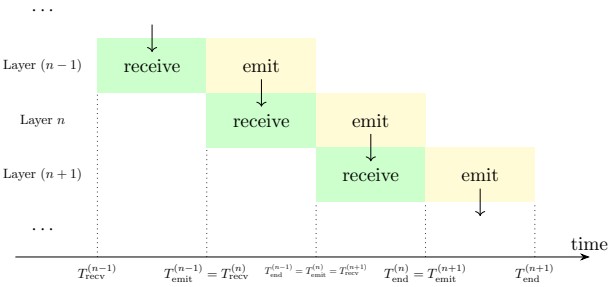

*Figure 1.* The pipeline of the receiving and emitting stages.

spikes from the previous layer, with its activation inhibited by setting a sufficiently large threshold. In the emitting stage, the neuron emits a spike to the next layer once the potential exceeds the threshold. Dividing the activation process into two stages effectively prevents the severe accuracy loss that occurs in a single-stage activation process, where early spike emissions from preceding neurons interfere with the complete processing of incoming information (Rueckauer & Liu, 2018).

An overview of the receiving and emitting process in the network is shown in Figure 1. For convenience, we assume that the receiving stage of layer $n$ corresponds exactly to the emitting stage of layer $(n+1)$. We denote the time range of the receiving stage as $[T_{\text{recv}}^{(n)}, T_{\text{emit}}^{(n)}]$ and the emitting stage as $[T_{\text{emit}}^{(n)}, T_{\text{end}}^{(n)}]$ in the $n$-th layer. Thus, $T_{\text{recv}}^{(n)} = T_{\text{emit}}^{(n-1)}$ and $T_{\text{end}}^{(n)} = T_{\text{emit}}^{(n+1)}$. The membrane potential dynamics of the $i$-th neuron in layer $n$ are given by (Stanojevic et al., 2023; 2024):

$$\frac{\mathrm{d}}{\mathrm{d}t}V_i^{(n)} = \begin{cases} A_i^{(n)} + \sum_j w_{ij}^{(n)} \mathcal{H}(t - t_j^{(n-1)}) \\ \qquad\qquad\qquad t \in [T_{\text{recv}}^{(n)}, T_{\text{emit}}^{(n)}), \\ B_i^{(n)} \qquad\qquad t \in [T_{\text{emit}}^{(n)}, T_{\text{end}}^{(n)}), \end{cases}$$
(5)

where $A_i^{(n)}$ and $B_i^{(n)}$ are predefined constants, $w_{ij}^{(n)}$ represents the synaptic weight, and $t_j^{(n-1)}$ denotes the spike time of the $j$-th presynaptic neuron.

Despite TTFS-based ANN-to-SNN conversion methods have been explored, their application to Transformers remains limited. This limitation arises mainly from two aspects.

First, TTFS-based neuron is limited by its representational range. Using the widely adopted mapping of spike time to values defined in Equation (1), the spike time can only represent values in the range of $[0, 1]$, which works well for ReLU-based networks. However, Transformer architectures often use complex activation functions like GELU (Hendrycks & Gimpel, 2016) and SiLU (Elfwing et al., 2018), which retain negative values and offer better performance than ReLU. Extending the representational range is necessary for effective TTFS-based conversion.

Secondly, the existing TTFS-based conversion methods struggle to handle the attention mechanism and Layer Norm in Transformers. The attention mechanism relies on the softmax operator, which is nonlinear, rendering existing TTFS-based methods inapplicable. Additionally, no TTFS-based implementation of Layer Norm has been proposed in previous studies. Addressing these challenges is essential for adapting TTFS-based SNNs to Transformer models.

## 4. Method: TTFSFormer

In this section, we propose generalized nonlinear neurons for converting Transformer activation functions. Using these neurons, we construct key Transformer layers and outline the complete conversion process, which we refer to as TTFSFormer.

### 4.1. Neuronal Dynamics in TTFSFormer

To overcome challenges in effectively representing nonlinear activations and accommodating extended value ranges, we incorporate two flexible kernel functions in the dynamics of TTFSFormer neurons and introduce the zero reference time mechanism.

With the initial value $V_i^{(n)}(T_{\text{recv}}^{(n)}) = 0.$, the change of the potential of the neuron in layer $n$ satisfies the following equation:

$$\frac{\mathrm{d}}{\mathrm{d}t}V_i^{(n)} = \begin{cases} \frac{1}{\tau^{(n)}} \cdot \left( \sum_j w_{ij}^{(n)} \eta_{ij}^{(n)}(t - t_j^{(n-1)}) + C_i^{(n)} \right) \\ \qquad\qquad\qquad t \in [T_{\text{recv}}^{(n)}, T_{\text{emit}}^{(n)}), \\ \psi_i^{(n)}(t - T_{\text{emit}}^{(n)}) \qquad t \in [T_{\text{emit}}^{(n)}, T_{\text{end}}^{(n)}), \end{cases}$$
(6)

where $w_{ij}^{(n)}$ are the weights of the $n$-th layer, $t_j^{(n-1)}$ is the spike time of the $j$-th input, the **input transform kernel** $\eta_{ij}^{(n)}$ is a function satisfying $\eta_{ij}^{(n)}(s) = 0, \forall s < 0$, $C_i^{(n)}$ is a constant, the **output transform kernel** $\psi_i^{(n)}$ is a non-negative function, and $\tau^{(n)}$ is the time constant of the $n$-th layer.

The relation between the spike time $t_i^{(n)} \in [T_{\text{emit}}^{(n)}, T_{\text{end}}^{(n)})$ in the $n$-th layer of SNN and the corresponding activation value $x_i^{(n)}$ in the $n$-th layer of ANN is

$$x_i^{(n)} \tau^{(n)} = T_{\text{ref}}^{(n)} - t^{(n)},$$
(7)

where the **zero reference time** $T_{\text{ref}}^{(n)}$ is manually selected. A spike at time $T_{\text{ref}}^{(n)}$ represents 0, and a larger value will result in an earlier spike. For simplicity, in the following discussion we denote $\delta^{(n)} = T_{\text{end}}^{(n)} - T_{\text{emit}}^{(n)}$ as the **time step**

of layer $n$. We assume the length of the time step is the same across all layers.

By modifying $T_{\text{ref}}^{(n)}$ and $\tau^{(n)}$, we can adjust the representational range $[a^{(n)}, b^{(n)}]$ in layer $n$ flexibly, where $a^{(n)} = \frac{T_{\text{ref}}^{(n)} - T_{\text{end}}^{(n)}}{\tau^{(n)}}$ and $b^{(n)} = \frac{T_{\text{ref}}^{(n)} - T_{\text{emit}}^{(n)}}{\tau^{(n)}}$. We denote the difference between these bounds as $d^{(n)} = b^{(n)} - a^{(n)}$.

## 4.2. Representational Ability of TTFSFormer Neurons

We theoretically prove that TTFSFormer neurons possess the representational capacity to convert Transformer architectures into SNNs by modifying the input and output transformation kernels $\eta, \psi$.

### 4.2.1. INPUT TRANSFORM

$V_i^{(n)}(T_{\text{emit}}^{(n)})$ can be regarded as a linear combination of transformed input values. The coefficient of combination is determined by $w$ while the transformation for input values is determined by $\eta$. Here is a generalized way to construct $\eta$ for a given transformation $f$.

**Theorem 4.1.** *Let $f_{ij} : [a^{(n-1)}, b^{(n-1)}] \rightarrow \mathbb{R}$ be differentiable functions. If we let*

$$\eta_{ij}^{(n)}(s) = \begin{cases} f'_{ij}\left(\frac{s}{\tau^{(n-1)}} + a^{(n-1)}\right) & s \geq 0 \\ 0 & s < 0 \end{cases}$$

$$C_i^{(n)} = \sum_j w_{ij} \frac{f_{ij}(a^{(n-1)})}{d^{(n-1)}}, \tag{8}$$

*then the potential at time $T_{\text{emit}}^{(n)}$ is*

$$V_i^{(n)}(T_{\text{emit}}^{(n)}) = \sum_j w_{ij}^{(n)} f_{ij}(x_j^{(n-1)}), \tag{9}$$

We call the potential at time $T_{\text{emit}}^{(n)}$ of a neuron in layer $n$ as its **accumulated potential**. Previous works use identity input transforms, which is a special version of Theorem 4.1.

**Corollary 4.2.** *In particular, if we let*

$$\eta_{ij}(s) = \mathcal{H}\left(\frac{s}{\tau^{(n-1)}} + a^{(n-1)}\right),$$

$$C_i^{(n)} = \sum_j \frac{a^{(n-1)}}{d^{(n-1)}} w_{ij}, \tag{10}$$

*Then*

$$V_i^{(n)}(T_{\text{emit}}^{(n)}) = \sum_j w_{ij}^{(n)} x_j^{(n-1)}. \tag{11}$$

*which is called an identity input transform.*

### 4.2.2. OUTPUT TRANSFORM

Now consider the relation between $V_i^{(n)}(T_{\text{emit}}^{(n)})$ and the output spike. For simplicity, we omit the subscript and

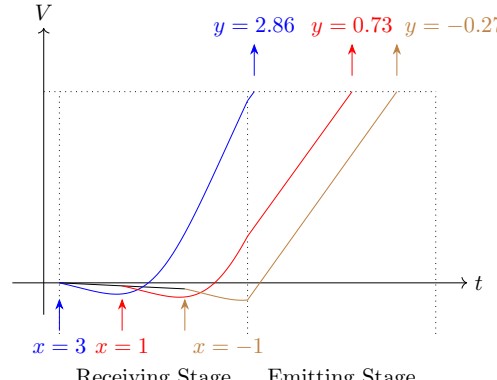

*Figure 2.* The dynamics of SiLU neuron, with input range $[-3, 3]$ and output range $[-1, 3]$. By Equation (16), in receiving stage, the potential increases and then decreases, and finally the rate of change tends to 1.

superscript since only one neuron is involved in emitting stage.

For simplicity, denote the clip function as

$$\text{clip}(x, a, b) = \begin{cases} a & x < a, \\ x & x \in [a, b], \\ b & x > b. \end{cases} \tag{12}$$

A general method to construct output transform by modifying $\psi$ is shown in Theorem 4.3.

**Theorem 4.3.** *Let $h : A \rightarrow \mathbb{R}$ be a differentiable monotone increasing function, and its inverse $h^{-1}$ is well-defined on $(a, b]$. If we let*

$$\psi(s) = \frac{1}{\tau h'\left(h^{-1}\left(b - \frac{s}{\tau}\right)\right)}, s \in [0, \delta),$$

$$\theta = h^{-1}(b), \tag{13}$$

*then the value $x$ represented by the output spike is*

$$x = \text{clip}(h(V(T_{\text{emit}})), a, b). \tag{14}$$

Previous works adopt an identity output transform, which is a special version of Theorem 4.1.

**Corollary 4.4.** *In particular, if we let $\psi(s) = \frac{1}{\tau}, \theta = b$, then the value $x$ represented by the output spike is*

$$x = \text{clip}(V(T_{\text{emit}}), a, b). \tag{15}$$

*which is called an identity output transform.*

The identity output transform is equivalent to what is used in previous work.

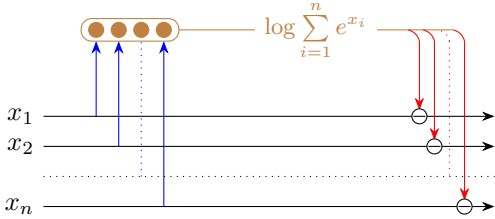

*Figure 3.* Structure of softmax operator for $x_1, \cdots, x_n$ (black lines). The brown part is the log-sum-exp neuron, with the blue arrows as its input and red arrows as its output.

### 4.3. Constructing TTFSFormer Layers

The Transformer architecture involves several non-linear layers. In this part, we will design their construction using spiking neurons.

#### 4.3.1. ACTIVATIONS

Every activation function can be implemented in the input transform stage according to Theorem 4.1. SiLU and GELU are two widely used activations in Transformer architectures, which can be constructed in TTFSFormer using the dynamics below.

**Corollary 4.5** (Construction of SiLU/GELU). *A neuron with input range $[a, b]$, time step $\delta$ and the identity output transform in Corollary 4.4 can represent* SiLU *function with*

$$\eta(s) = \mathbb{I}[s \geq 0] \cdot \sigma(u) \cdot (u + 1 - u \cdot \sigma(u)),$$
$$\text{where } u = a + \frac{\delta}{b-a}s, \ \sigma(x) = \frac{1}{1 + e^{-x}}, \tag{16}$$

*or* GELU *function with*

$$\eta(s) = \mathbb{I}[s \geq 0] \cdot \left[ \frac{1}{2} + \frac{\operatorname{erf}(\frac{u}{\sqrt{2}})}{2} + \frac{u}{\sqrt{2\pi}} e^{-\frac{u^2}{2}} \right],$$
$$\text{where } u = a + \frac{\delta}{b-a}s, \ \operatorname{erf}(x) = \frac{2}{\sqrt{\pi}} \int_0^x e^{-t^2} \, \mathrm{d}t. \tag{17}$$

A full example of SiLU neurons is shown in Figure 2.

#### 4.3.2. SOFTMAX

Softmax is another important operation in the attention mechanism. The softmax of a row vector $(x_1, \cdots, x_n)$ is defined as

$$\text{Softmax}(x_1, \cdots, x_n) = \left( \frac{e^{x_1}}{\sum_j e^{x_j}}, \cdots, \frac{e^{x_n}}{\sum_j e^{x_j}} \right). \tag{18}$$

Firstly, we construct the log-sum-exp neuron.

**Theorem 4.6.** *The log-sum-exp of $n$ inputs $x_1, x_2, \cdots, x_n$,*

*i.e. $\log \sum_{i=1}^n e^{x_i}$, can be calculated in a single neuron with*

$$\eta^{(1)}(x) = \exp\left( \frac{s}{\tau^{(0)}} + a^{(0)} \right)$$
$$C^{(1)} = \frac{n}{d^{(0)}} e^{a^{(0)}} \tag{19}$$
$$\psi^{(1)}(s) = \frac{1}{\tau^{(1)}} \exp\left( b - \frac{s}{\tau^{(1)}} \right)$$

*where the current layer is layer* 1.

With the log-sum-exp neuron, we can obtain the softmax operator. We can calculate the logarithm of softmax, i.e.

$$\log \frac{e^{x_i}}{\sum_{j=1}^n e^{x_j}} = x_i - \log \sum_{j=1}^n e^{x_j}, \tag{20}$$

by subtracting the log-sum-exp from $x_i$. Finally, we can obtain the output after an exponential layer. The whole process is shown in Figure 3.

#### 4.3.3. LAYERNORM

LayerNorm is a normalization method widely used in transformer architecture, achieving better results than BatchNorm, which is a linear operation. LayerNorm is defined as

$$\text{LayerNorm}(x) = \frac{x - \mathbb{E}(x)}{\sqrt{\text{Var}(x) + \varepsilon}} \cdot \gamma + \beta. \tag{21}$$

The LayerNorm operator can be obtained by the following parts. Firstly, the mean $\bar{x}$ can be calculated directly by a single neuron, after which we subtract $\bar{x}$ from each $x_i$. Then, we can obtain the variance $\text{Var}(x)$ by a single neuron with

$$\eta^{(1)}(s) = 2\left( \frac{s}{\tau^{(0)}} + a^{(0)} \right)$$
$$C^{(1)} = \frac{(a^{(0)})^2}{nd^{(0)}} \quad w_i^{(1)} = \frac{1}{n} \tag{22}$$

and identity output transform. After that, we can get $\frac{1}{\sqrt{\text{Var}+\varepsilon}}$ by single neuron with

$$\eta^{(2)}(s) = \frac{1}{2}\left( \frac{s}{\tau^{(1)}} + a^{(1)} \right)^{-\frac{3}{2}}$$
$$C^{(2)} = \frac{1}{d^{(1)}\sqrt{a^{(1)} + \varepsilon}} \quad w^{(2)} = 1 \tag{23}$$

and identity output transform. Finally, multiply $x_i$ with $\frac{1}{\sqrt{\text{Var}+\varepsilon}}$, which is discussed in Section 4.3.4.

#### 4.3.4. MULTIPLICATION

Multiplication is one of the core operations in the attention mechanism of Transformer. We first consider the multiplication of two neurons $x_1, x_2 \in [a^{(0)}, b^{(0)}]$, with which we can implement matrix product and pointwise product directly.

*Table 1.* Comparison between our method and previous works on ImageNet1k dataset

| Work | Method | Architecture | Param. | SNN Acc. | ANN Acc. |
|---|---|---|---|---|---|
| Spikingformer (Zhou et al., 2023a) | Direct Training | Spikingformer-8-512
Spikingformer-8-768 | 29.68M
66.34M | 74.79%
75.85% | -
- |
| Spike-Driven (Yao et al., 2023; 2024) | Direct Training | SpikeDriven-V1
SpikeDriven-V2 | 66.34M
55.4M | 77.07%
80.0% | -
- |
| E-Spikeformer (Yao et al., 2025) | Direct Training | E-Spikeformer
E-Spikeformer | 83.0M
173.0M | 85.2%
86.2% | -
- |
| SRP (Hao et al., 2023) | CNN-to-SNN | ResNet-34
VGG-16 | 21.8M
138M | 68.61%
69.43% | 74.32%
74.29% |
| Lossless TTFS (Stanojevic et al., 2023; 2024) | CNN-to-SNN (TTFS-based) | ResNet-34
VGG-16 | 21.8M
138M | 75.36%
75.66% | 74.32%
74.29% |
| MST (Wang et al., 2023) | Transformer-to-SNN | Swin-T(BN) | 28.5M | 78.51% | 89.51% |
| STA (Jiang et al., 2024) | Transformer-to-SNN | ViT-B/32 | 86M | 82.79% | 85.10% |
| ECMT (Huang et al., 2024) | Transformer-to-SNN | ViT-L/16
EVA-G | 307M
1074M | 84.60%
88.60% | 85.83%
88.88% |
| SpikeZIP-TF (You et al., 2024) | Transformer-to-SNN | ViT-S/16
ViT-B/16
ViT-L/16 | 22M
86M
307M | 81.45%
82.71%
83.42% | 81.38%
85.10%
85.83% |
| AdaFire (Wang et al., 2025) | Burst Coding | ViT-S/16
ResNet-34 | 22M
21.8M | 77.09%
75.38% | 81.38%
74.32% |
| **TTFSFormer (Ours)** | Transformer-to-SNN (TTFS-based) | ViT-S/16
ViT-B/16
ViT-L/16
EVA-G
EVA02-S
EVA02-L | 22M
86M
307M
1074M
22.1M
305M | 81.40%
85.07%
85.78%
88.90%
85.68%
90.03% | 81.38%
85.10%
85.83%
88.88%
85.72%
90.05% |

Now we construct a two-layer network that computes their product $x_1 \cdot x_2$.

**Layer I**. The neuron on layer I takes $x_1, x_2 \in [a^{(0)}, b^{(0)}]$ as its input value. Let constant $c = -a^{(0)} + 1$. By Theorem 4.1, with

$$\eta_i^{(1)}(s) = \frac{1}{\frac{s}{\tau^{(0)}} + a^{(0)} + c} = \frac{1}{1 + \frac{s}{\tau^{(0)}}} \quad (24)$$
$$C_i^{(1)} = 0 \quad w_i^{(1)} = 1$$

the accumulated potential is

$$V^{(1)}(T_{\text{emit}}^{(1)}) = \log(x_1 + c) + \log(x_2 + c) \\ = \log(x_1 x_2 + c x_1 + c x_2 + c^2) \quad (25)$$

By taking the identity output transform illustrated in Corollary 4.4, the neuron produces the output $y = \log(x_1 x_2 + c x_1 + c x_2 + c^2)$.

**Layer II**. The neuron on layer II takes $y, x_1, x_2$ as its input

value. By Theorem 4.1, with

$$\eta_0^{(2)}(s) = \exp\left(\frac{s}{\tau^{(1)}} + a^{(1)}\right) \quad (26)$$

for the input $y$, the identity input transform settings illustrated in Corollary 4.2 for input $x_1, x_2$ and

$$C_0^{(2)} = \frac{1}{d^{(1)}} e^{a^{(1)}} + \frac{2 c a^{(1)}}{d^{(1)}} \\ w_0^{(2)} = 1 \qquad w_1^{(2)} = w_2^{(2)} = -c \quad (27)$$

for $y, x_1, x_2$ respectively, the accumulated potential is $V^{(2)}(T_{\text{emit}}^{(2)}) = x_1 x_2 + c^2$. In the output transform, by letting $\psi^{(2)}(s) = \frac{1}{\tau^{(2)}}$ and $\theta^{(2)} = b^{(2)} - c^2$, we get the final output representing $x_1 x_2$.

## 5. Experiments

In this section, we evaluate our TTFS-based converted SNN methods on the ImageNet-1k dataset (Deng et al., 2009) and

compare with the corresponding ANNs and other previously proposed SNN methods. Moreover, we estimate the energy efficiency and explore the robustness of our method.

We test the conversion method on various architectures with different model sizes, including ViT (ViT-S, ViT-B, ViT-L) (Dosovitskiy et al., 2020; Steiner et al., 2022) and EVA (EVA-S, EVA-L) (Fang et al., 2023; 2024). The weights are sourced from public repositories on Hugging Face. Using pre-trained ANN models, SNNs are obtained through Algorithm 1 in the supplementary material. The detailed results are presented in Table 1, along with comparisons to state-of-the-art SNN methods.

### 5.1. Experimental Results

Generally speaking, our method achieve comparable performance to the original ANN architecture, with less than 0.1% drop in accuracy. Specifically, we achieve accuracies of 85.8% and 90.0% for the ViT-L/16 and EVA-L architectures respectively. Furthermore, the results of these large models demonstrate the superior scalability of our approach.

Our work outperforms previously proposed SNN Transformers. By leveraging pre-trained ANN models, we achieve significantly higher accuracy than directly trained SNN Transformers, while maintaining low computational cost.

Compared to rate-coding based SNN conversion methods on Transformer architectures, such as STA (Jiang et al., 2024), ECMT (Huang et al., 2024), and SpikeZIP-TF (You et al., 2024), our proposed method further bridges the gap between the accuracy of SNNs and the original ANN. SNNs obtained from rate-coding methods require multiple spikes to approximate the activation value, which creates a gap between the activation values of an ANN layer and the mean firing rates of the corresponding SNN layer, especially for non-linear layers. In contrast, the TTFS encoding method conveys the output of a layer in a single spike, allowing the next layer to receive complete information as soon as it receives spikes.

Compared to previous TTFS-based conversion methods (Stanojevic et al., 2023; 2024), we successfully integrate Transformer architectures into SNNs by utilizing the proposed neuron model and achieve significantly better results than TTFS-based convolutional SNNs.

### 5.2. Energy Estimation

Since in TTFS coding, the input activation is encoded into one spike before it reaches the first layer. Consequently, there is no need to perform multiplication operations in the first layer. The energy cost can then be expressed using the

*Table 2.* Energy consumption for different architectures

| Architecture | $OP_{\mathrm{SNN}}$ ($\times 10^9$) | $E_{\mathrm{SNN}}$ | $\frac{E_{\mathrm{SNN}}}{E_{\mathrm{ANN}}}$ |
|---|---|---|---|
| ViT-S/16 | 5.42 | 4.9mJ | 22.3% |
| ViT-B/16 | 19.2 | 17mJ | 21.0% |
| ViT-L/16 | 65.7 | 59mJ | 20.7% |

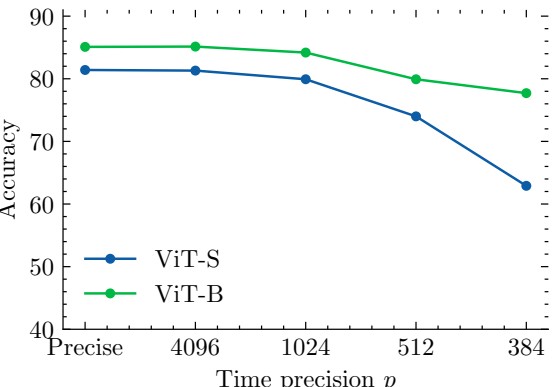

*Figure 4.* Accuracy of ViT under different time precision of TTFS equation provided in (Rathi & Roy, 2023):

$$\frac{E_{\mathrm{SNN}}}{E_{\mathrm{ANN}}} = \frac{OP_{\mathrm{SNN}} \cdot E_{\mathrm{AC}}}{OP_{\mathrm{ANN}} \cdot E_{\mathrm{MAC}}} \tag{28}$$

The operation counts $OP_{\mathrm{SNN}}$ are summarized in Table 3 in Appendix C. We set $E_{\mathrm{AC}} = 0.9\mathrm{pJ}$ and $E_{\mathrm{MAC}} = 4.6\mathrm{pJ}$ according to (Horowitz, 2014). The calculated energy consumption is shown in Table 2. The results indicates that our method achieves comparable performance to the original ANN architecture with only about 20% energy consumption.

### 5.3. Robustness

In previous sections, the emitting and receiving processes are simulated by passing the time-to-first-spike across different layers using floating-point values for TTFS. However, due to the limitation of hardware implementation, the accuracy of TTFS can hardly be as accurate as 32-bit floating-point numbers. Thus, we explore the robustness of our implementation in this section.

In hardware implementation, we define the time precision $p$ of TTFS as the ratio of the time step $\delta$ to the maximum error $\epsilon_{\max}$ of the simulated values for TTFS, given by $p = \frac{\delta}{\epsilon_{\max}}$. We test the accuracy loss of ViT models under different TTFS time precision, as shown in Figure 4. Our method can obtain comparable performance to the original ANN without fine-tuning at a time precision of 1024 or higher. However, the performance declines with decreasing time precision, which requires a trade-off between latency and

accuracy.

## 6. Conclusion

In this paper, we propose a novel conversion method of the transformer architecture with TTFS-coding, which is the first work on TTFS-based spiking transformer to our best knowledge. With the proposed structure of generalized neurons in TTFSFormer, our method is able to convert transformer architecture into SNN with little accuracy loss. Taking full advantage of the energy efficiency of SNN, our method achieves a comparable accuracy to the original ANN architecture with much lower energy cost.

The TTFS-coding demonstrates both strong representational capability in representing information and perfect energy efficiency. The TTFS-coding method might be a promising direction for future works. While our method focuses on ANN-to-SNN conversion, future works can explore a training framework for TTFS-coding transformer SNN.

## Acknowledgments

This work was supported by the National Natural Science Foundation of China (62422601, U24B20140, and 62088102), Beijing Municipal Science and Technology Program (Z241100004224004), Beijing Nova Program (20230484362, 20240484703), and National Key Laboratory for Multimedia Information Processing.

## Impact Statement

This paper presents work whose goal is to advance the field of Machine Learning. There are many potential societal consequences of our work, none which we feel must be specifically highlighted here.

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

---

**Algorithm 1** Converting ANN into TTFS-based SNN

---

    **Input:** ANN $P$.

    Select the time step $\delta$ according to hardware.

    Set the constants $T_{\text{emit}}^{(n)} = T_{\text{emit}}^{(0)} + n\delta$ and $T_{\text{end}}^{(n)} = T_{\text{emit}}^{(n)} + \delta$, where $T_{\text{emit}}^{(0)}$ is the tick when simulation begins.

    **for** layer $L_n$ **in** $P$ **do**

        Monitor the range of output $X_n = [a_n, b_n]$.

        Select the time constant as $\tau^{(n)} = \frac{\delta}{b_n - a_n}$.

        Set $T_{\text{ref}}^{(n)} = T_{\text{emit}}^{(n)} + b_n \tau^{(n)}$.

        **if** $L_n$ is `Softmax` or $L_n$ is `LayerNorm` **then**

            Simulate the spiking counterpart illustrated in Sections 4.3.2 and 4.3.3.

            Monitor the range of the output in each intermediate layer.

        **end if**

    **end for**

    **for** layer $L_n$ **in** $P$ **do**

        Convert $L_n$ into a Spiking Layer according to Section 4.3.

    **end for**

---

# A. Experiment Details

The whole conversion process is shown in Algorithm 1. We'll discuss some details in this part.

## A.1. Setting the Constants

Since we're using adjustable parameters $\tau$ and $T_{\text{ref}}$, we can set the $[a, b]$ such that nearly all outputs lie within the range. More specifically, if the output range is $[a, b]$, we can set

$$
\begin{aligned}
b\tau &= T_{\text{ref}} - T_{\text{emit}}, \\
a\tau &= T_{\text{ref}} - T_{\text{end}},
\end{aligned}
\tag{29}
$$

which indicates that $\tau = \frac{\delta}{b-a}$ and $T_{\text{ref}} = T_{\text{emit}} + b\tau$.

## A.2. Evaluating

Evaluating the SNN is similar to ANN except for the conversion between spikes and values. Since the network takes spikes as its input, a pixel value $x \in \mathbb{R}$ is turned to a spike at time $t = T_{\text{ref}}^{(0)} - x\tau$. Besides, as the output of the network are actually spikes, the spikes should be turned back into real values. However, in image classification, the prediction is given by the index of the earliest spike, and there is no need to convert the spike back into real values.

# B. Proof of Theorem

## B.1. Proof of Theorem 4.1

*Proof.* Consider the potential change in the receiving stage.

$$
\begin{aligned}
V_i^{(n)}(T_{\text{emit}}^{(n)}) &= \frac{1}{\tau} \int_{T_{\text{emit}}^{(n-1)}}^{T_{\text{end}}^{(n-1)}} \sum_j w_{ij}^{(n)} \eta_{ij}^{(n)}(t - t_j^{(n-1)}) + C_i^{(n)} \, \mathrm{d}t \\
&= \frac{1}{\tau} \sum_j w_{ij}^{(n)} \int_0^{T_{\text{end}}^{(n-1)} - t_j^{(n-1)}} \eta_{ij}^{(n)}(s) \, \mathrm{d}s + d^{(n-1)} \cdot C_i^{(n)} \\
&= \sum_j w_{ij}^{(n)} \frac{1}{\tau^{(n-1)}} \int_0^{T_{\text{end}}^{(n-1)} - T_{\text{ref}}^{(n-1)} + \tau^{(n-1)} x_j^{(n-1)}} \eta_{ij}^{(n)}(s) \, \mathrm{d}s + d^{(n-1)} \cdot C_i^{(n)} \\
&= \sum_j w_{ij}^{(n)} \, f_{ij} \left( \frac{s}{\tau^{(n-1)}} + a^{(n-1)} \right) \Big|_0^{\tau^{(n-1)}(x_j^{(n-1)} + a^{(n-1)})} + d^{(n-1)} \cdot C_i^{(n)} \\
&= \sum_j w_{ij}^{(n)} \left( f_{ij}(x_j^{(n-1)}) - f_{ij}(a^{(n-1)}) \right) + d^{(n-1)} \cdot C_i^{(n)} \\
&= \sum_j w_{ij}^{(n)} f_{ij}(x_j^{(n-1)})
\end{aligned}
\tag{30}
$$

$\square$

## B.2. Proof of Theorem 4.3

*Proof.* Denote $g = h^{-1}$. Since $h(g(x)) = x$, we have

$$
h'(g(x)) \cdot g'(x) = 1
\tag{31}
$$

by taking the derivative of both side.

If the spike is emitted at time $t \in [T_{\text{emit}}, T_{\text{end}})$, i.e. the corresponding value $x \in (a, b]$. Then

$$
\begin{aligned}
\theta &= V(T_{\text{emit}}) + \int_0^{t - T_{\text{emit}}} \psi(s) \, \mathrm{d}s \\
&= V(T_{\text{emit}}) + \int_0^{T_{\text{ref}} - T_{\text{emit}} - \tau x} \frac{1}{\tau} g' \left( b - \frac{s}{\tau} \right) \, \mathrm{d}s \\
&= V(T_{\text{emit}}) - g \left( b - \frac{s}{\tau} \right) \Big|_0^{\tau(b-x)} \\
&= V(T_{\text{emit}}) - g(x) + g(b)
\end{aligned}
\tag{32}
$$

Thus

$$
g(x) = V(T_{\text{emit}})
\tag{33}
$$

which indicates that

$$
x = h(V(T_{\text{emit}})) \in (a, b]
\tag{34}
$$

If $h(V(T_{\text{emit}})) > b$, then $V(T_{\text{emit}}) > g(b) = \theta$, which means that a spike is emitted once the emitting window begins, i.e. at $T_{\text{emit}}$, representing the value $\frac{T_{\text{ref}} - T_{\text{emit}}}{\tau} = b$. If $h(V(T_{\text{emit}})) < a$, then the potential at time $T_{\text{end}}$ is (according what we calculated before)

$$
V(T_{\text{emit}}) + \int_0^{\delta} \psi(s) \, \mathrm{d}s = V(T_{\text{emit}}) - g \left( b - \frac{s}{\tau} \right) \Big|_0^{\delta} = V(T_{\text{emit}}) - g(a) + g(b) < g(b) = \theta
\tag{35}
$$

which means that there will be no spike, representing the value $a$. $\square$

*Table 3.* Energy consumption for all layers

| Layer | Operand Size | Number of Neurons | Operation Counts |
|---|---|---|---|
| $\mathrm{MatMul}(A, B)$ | $A \in \mathbb{R}^{M \times K}$ and $B \in \mathbb{R}^{K \times N}$ | $MN(K + 1)$ | $MN(3K + 2)$ |
| $\mathrm{Softmax}(x)$ | $x \in \mathbb{R}^N$ | $2(N + 1)$ | $6N$ |
| $\mathrm{LayerNorm}(x)$ | $x \in \mathbb{R}^N$ | $3N + 3$ | $9N + 1$ |
| Activation (GELU/SiLU) $f(A)$ | $A \in \mathbb{R}^{M \times N}$ | $MN$ | $MN$ |

# C. Detailed Construction of Operators

In this part, we give detailed construction of the operators mentioned before, along with some less important optimization techniques.

## C.1. Multiplication

The product of two neurons can be calculated by a two-layer network, as described in Section 4.3.4.

In the multiplication of matrix $A \in \mathbb{R}^{M \times K}$ and matrix $B \in \mathbb{R}^{K \times N}$, we need $MNK$ neurons in the first layer, each taking $A_{ik}, B_{kj}$ as input for $1 \le i \le M, 1 \le k \le K, 1 \le j \le N$. In the second layer, we should combine the neuron that shares the same $k$, i.e. each neuron takes $\{A_{ik}, B_{kj}, y_{ikj} | 1 \le k \le K\}$ as input for $1 \le i \le M, 1 \le j \le N$. The threshold should be changed into $\theta^{(2)} = b^{(2)} - Kc^2$.

## C.2. Softmax

Consider $n$ neurons with input $x_1, x_2, \cdots, x_n \in [a^{(0)}, b^{(0)}]$.

**Layer 0** (Optional). The result of softmax wouldn't change if we subtract a constant for all $x_i$, which can improve the numerical stability of softmax. If we choose to subtract $\max_i x_i$, we may need to add a layer 0.

More specifically, we can add a "maximum" neuron taking $x_1, \cdots, x_n$ as its input, and emit a spike the first time it receives any input spike. The neuron represents the maximum value among all $x_i$. Layer 0 contains $n$ neurons in addition to the maximum neuron, where the $i$-th neuron do the subtraction of $x_i$ and $\max_j x_j$.

**Layer I**. The first layer contains 1 neuron described in Theorem 4.6, whose output is $y = \log \sum_i e^{x_i}$.

**Layer II**. The second layer contains $n$ neurons, where the $i$-th neuron takes $x_i$ and $y$ as its input. With identity input transform and weight 1 for $x_i$ and $-1$ for $y$, we can get the accumulated potential $x_i - y$. With

$$h^{(2)}(x) = e^x$$
$$\psi^{(2)}(s) = \frac{1}{T_{\mathrm{ref}} - T_{\mathrm{emit}} - s} \tag{36}$$
$$\theta^{(2)} = \log b^{(2)}$$

We can get the output

$$\exp\left(x_i - \log \sum_{j=1}^n e^{x_j}\right) = \frac{e^{x_i}}{\sum_{j=1}^n e^{x_j}} \tag{37}$$

The result of softmax function must be in $(0, 1)$, so we can set $a^{(2)} = 0, b^{(2)} = 1$.

