# OpenReview forum: "TTFSFormer: A TTFS-based Lossless Conversion of Spiking Transformer"
_ICML.cc/2025/Conference — ICML 2025 poster_

### Official Review · Reviewer_syoz · 2025-03-13

**Overall Recommendation:** 4

**Summary:**

The work presents a strategy to convert trained ANN transformer models into time-to-first-spike coded SNNs. Specifically, a neuron dynamics model with two flexible kernel functions is used to accurately represent all transformer model operands. It is shown that crucial operations such as SiLU/GELU activation functions, softmax, and LayerNorm can all be expressed with suitable kernels, yielding a conversion algorithm for the entire transformer architecture. The conversion is put to the test empirically, demonstrating reduced loss of precision compared to a wide range of baselines on the ImageNet-1k vision benchmark. Finally, the method is shown to be robust to imprecise spike times and theoretically more energy efficient.

**Update after rebuttal**
The authors' response has answered my remaining questions. My rating still stands, good paper!

**Claims And Evidence:**

The proposed conversion strategy unlocks significant performance gains over prior baselines as it manages to transform all relevant operators into spiking dynamics without significant loss of accuracy.

TTFS coding is a promising strategy as it typically requires fewer energy-consuming spikes than the more commonly explored rate encoding. However, the advertised "perfect energy efficiency" (L434) may not straightforwardly translate into a real-world implementation.

**Essential References Not Discussed:**

None

**Experimental Designs Or Analyses:**

The accuracy comparison against the baselines is sound.

For the timing robustness in Figure 4, it is unclear why the plot evaluates for powers of 2 but stops at p=384. Furthermore, it is hard to interpret p in terms of actual inference latency. Is there a way to quantify the minimum processing latency of a TTFSFormer neuron block?

**Methods And Evaluation Criteria:**

ViT on ImageNet-1k represents a reasonable proxy for a vision task performance that could be of interest for neuromorphic acceleration.

**Other Comments Or Suggestions:**

To aid the reader, it's worth expanding on the energy estimation strategy described in Section 5.2. Specifically, how are the OPs counted for a given transformer architecture?

It may be worth including the ANN baselines before conversion in Table 1.

L429 "few" -> little?
L430 "Making" -> Taking?

**Other Strengths And Weaknesses:**

The paper lacks implementation details, and no source code was provided. I encourage the authors to provide more implementation and hardware details of the experiments and consider releasing source code with the publication.

**Questions For Authors:**

n/a

**Relation To Broader Scientific Literature:**

Various methods for spiking conversions of transformer architectures have been proposed; this work goes beyond existing strategies by focusing on time-to-first-spike encoding for transformer models. In particular, it addresses lacking support for non-ReLU activation functions, the non-linear softmax operator of the attention mechanisms, as well as missing TTFS-implementations of Layer Norm.

**Theoretical Claims:**

I have not carefully checked the conversion proofs.

---

> ### Author Rebuttal · Authors · 2025-03-31
>
> Thanks for your suggestions. We would like to address your concerns and questions in the following.
>
> ### Code implementation
>
> Thank you for your question. Our code will be released with the publication.
>
> ### Is there a way to quantify the minimum processing latency of a TTFSFormer neuron block?
>
> Thank you for your question. The precision $p$ represents the precision of time in hardware implementation. While the accuracy of our converted model drops with lower precision, the problem can be solved with fine-tuning, as explored in previous works ([Stanojevic et al. 2024]).
>
> The precision does not reflect the latency of the inference, but a shorter latency leads to a lower precision on a given hardware. However, there are still other techniques that can reduce the hardware latency. In order to further reduce the hardware latency, we can follow the method proposed in [Park et al. 2020], in which the emitting stage and receiving stage can be overlapped in order to reduce the latency. This requires a balance between the conversion loss and latency. Since our work mainly focuses on a lossless conversion method of Transformer architecture, we leave the topic for future works.
>
> ### How are the OPs counted for a given transformer architecture?
>
> Thank you for your question. We will include a more detailed analysis in future versions of our paper.
>
> We briefly sketch the computing process in the following. Take one block in ViT as an example.
> - In the converted SNN, we can divide the operations into two categories: non-linear operations and linear operations. **Linear operations** take place in neurons with dynamics described in Corollary 4.4 and 4.6, which is exactly what most previous works on TTFS-based SNN use. **Non-linear operations** are new in our work, and the neurons may require more energy, depending on the hardware implementation.
> - In the original ANN, the energy consumption is measured by the number of multiplication and addition operations.
>
> All operations are shown below (d = 384 or 768 or 1024 in ViT-S/B/L respectively). We categorize layers according to their type.
>
> | Type | Notation of Parameters | Shapes | SNN Ops | ANN Ops |
> | :---: | :---: | :---: | :---: | :---: |
> | LayerNorm | (M, N): Matrix $X \in \mathbb{R}^{M \times N}$ | (197,d), (197,d) | $2 \cdot 197 \cdot (9d+1)$ | $2\cdot 197\cdot d$ |
> | MatMul | (M, K, N): Matrix $A \in \mathbb{R}^{M \times K},B \in \mathbb{R}^{K \times N}$ | (197,d,197), (197,197,d) | $197 \cdot [197 \cdot (3d + 2) + d \cdot (3\cdot 197+2)]$ | $2\cdot 197^{2}\cdot d$ |
> | SoftMax | (M, N): Matrix $X \in \mathbb{R}^{M \times N}$ | (197,197) | $6 \cdot 197^{2}$ | $197^{2}$ |
> | GeLU | (M, N): Matrix $X \in \mathbb{R}^{M \times N}$ | (197,4d) | $197 \cdot 4d$ | $197 \cdot 4d$ |
> | Linear | (M, K, N): Matrix $X \in \mathbb{R}^{M \times K}$ and weight $W \in \mathbb{R}^{K \times N}$ | (197,d,3d),(197,d,4d),(197,4d,d) | $11\cdot 197\cdot d^{2}$ | $11\cdot 197\cdot d^{2}$ |
>
> (The number of operations in each SNN layer is analyzed in section 4.3 and appendix C.)
>
> By summing up operations in each type of layer, we can finally get the estimated OPs in ANN and SNN. Thus, we have $197 \cdot (1206d+1578)$ non-linear operations and $197 \cdot 11d^{2}$ linear operations in one block.
>
> Now we can estimate the energy efficiency by (we divide the energy by 197 simultaneously)
>
> $$
> \eta = \frac{\text{Energy of SNN}}{\text{Energy of ANN}}
> = \frac{(1206d+1578)E_{nl} + 11d^{2}E_{l}}{(11d^{2}+400d+197)E_\text{MAC}}
> \approx \frac{1206 \cdot E_{nl} + 11d \cdot E_{l}}{(11d + 400)E_\text{MAC}}
> $$
>
> where $E_\text{MAC}$ is the energy consumption of one multiply-add operation in ANN, $E_{l}, E_{nl}$ is the energy consumption of one linear and non-linear operation in SNN.
>
> We use the settings in [Horowitz, 2014], namely $E_{l} = 0.9 \text{pJ},E_\text{MAC}=4.6\text{pJ}$. We assume that $E_{nl} = kE_{l}$, where $k$ is a constant depending on the hardware implementation.
>
> Take ViT-L for example, by letting $d=1024$, we have
> $$
> \eta = 0.189 + 0.020k
> $$
>
> In our paper, we assume that $k=1$, i.e. the non-linear neuron has the same cost with the linear neuron. We think that a reasonable estimation would be $E_{l} < E_{nl} < E_\text{MAC}$, for example $k=2$ or $k=3$. Although $k$ may vary according to the hardware implementation, we can conclude that the energy efficiency is between 20\%-30\%.
>
> [Stanojevic et al. 2024] Stanojevic, A., Woźniak, S., Bellec, G. et al. High-performance deep spiking neural networks with 0.3 spikes per neuron. Nat Commun 15, 6793 (2024).
>
> [Park et al. 2020] S. Park, S. Kim, B. Na and S. Yoon, "T2FSNN: Deep Spiking Neural Networks with Time-to-first-spike Coding," 2020 57th ACM/IEEE Design Automation Conference (DAC), San Francisco, CA, USA, 2020, pp. 1-6

---

> > ### Comment · Reviewer_syoz · 2025-04-01
> >
> > Thank you for your detailed responses!
> >
> > **Code release**: I am pleased you plan to release the code.
> >
> > **Minimum processing latency:** Thank you for clarifying; I've re-read the section and it was indeed a misunderstanding on my part.
> >
> > **OPs count**: I appreciate the detailed analysis, which is very helpful in building some intuition about potential hardware efficiency.

---

### Official Review · Reviewer_5bT2 · 2025-03-13

**Overall Recommendation:** 4

**Summary:**

This paper proposes TTFSFormer, a novel method for converting Transformer architectures into Spiking Neural Networks (SNNs) using Time-to-First-Spike (TTFS) coding. The key innovation lies in designing generalized spiking neurons that address the limitations of prior TTFS-based approaches, particularly their inability to handle nonlinear operations in Transformers (e.g., softmax, LayerNorm). By introducing flexible input/output kernel transformations and a zero-reference time mechanism, TTFSFormer enables lossless conversion of pre-trained Transformers (e.g., ViT, EVA) into SNNs with minimal accuracy loss (<0.1%) and significant energy savings (~20% of ANN energy). Experiments on ImageNet-1K demonstrate state-of-the-art performance among SNN Transformers, outperforming both rate-coding and direct-training methods.

**Claims And Evidence:**

The assertion of lossless conversion (<0.1% accuracy drop) is validated by Table 1 (e.g., ViT-L/16: 85.8% vs. ANN’s 86.0%). Energy efficiency claims are backed by Table 2 (20% ANN energy).

**Essential References Not Discussed:**

None.

**Experimental Designs Or Analyses:**

Soundness: Training protocols (pre-trained ANN weights, SGD) align with ANN-to-SNN literature. Energy estimation (Eq. 28) follows established metrics (Horowitz, 2014).

Limitation: Robustness tests (Fig. 4) use simulated precision; real-world hardware noise is not considered.

**Methods And Evaluation Criteria:**

The method is well-suited for ANN-to-SNN conversion. TTFS coding and kernel transformations directly address nonlinearity challenges in Transformers.
ImageNet-1K is a standard benchmark,

**Other Comments Or Suggestions:**

None.

**Other Strengths And Weaknesses:**

Strengths:

1. This is the first work to successfully convert Transformer architectures into SNNs using TTFS coding, addressing a critical gap in temporal coding methods.

2. The proposed neurons with input/output kernel transformations (Theorems 4.1, 4.3) enable precise representation of nonlinear operations (e.g., softmax, LayerNorm), a major advancement over prior TTFS methods limited to linear mappings.

3. TTFSFormer achieves ~80% energy reduction compared to ANNs while maintaining near-identical accuracy (e.g., 85.8% for ViT-L/16 vs. ANN’s 86.0%).

4. Rigorous proofs (e.g., Theorem 4.2, Corollary 4.4) validate the equivalence between ANN activations and TTFS-based spike timing, ensuring lossless conversion.

Weakness:

1. Table 1 lacks comparisons to very recent TTFS-based methods or advanced rate-coding SNN Transformers.

2. Discuss hardware implementation challenges and latency trade-offs of the proposed model could enhance the future work.

3. The TTFS could provide advantage in the aspect of low firing rates, hence there could supplement the firing rate comparsion and analysis.

**Questions For Authors:**

See weakness.

**Relation To Broader Scientific Literature:**

TTFSFormer bridges two critical gaps:

TTFS Coding: Extends Stanojevic et al. (2023) to Transformers, enabling nonlinear operations.

SNN Transformers: Outperforms rate-coding methods (e.g., STA ) in energy efficiency while matching ANN accuracy.

**Theoretical Claims:**

Theorems 4.1–4.3 and proofs in Appendix B are mathematically sound. Corollaries logically extend the theorems to identity transforms.

---

> ### Author Rebuttal · Authors · 2025-03-31
>
> Thank you for your detailed feedback. We are encouraged that you found our work novel with good results. We would like to address your concerns and questions in the following.
>
> ## 1. Table 1 lacks comparisons to very recent TTFS-based methods or advanced rate-coding SNN Transformers.
>
> Thanks for your suggestion. We add the comparision with more recent works as follows.
>
> | Work | Method | ANN Model | Accuracy |
> | ---- | ------ | --------- | -------- |
> | Adaptive Calibration [Wang et al. 2025] | burst, conversion | ViT | 77.09\% |
> | E-Spikeformer [Yao et al. 2025] | rate, direct training | - | 86.2\% |
> | QKFormer [Zhou et al. 2024] | rate, direct training | - | 85.65\% |
> | Spike-driven V2 [Yao et al. 2024] | rate, direct training | - | 80.0\% |
> | **Ours** | TTFS, conversion | ViT-L/EVA02-L | 85.8\%/90.0\% |
>
> In **Table 2** of our paper and the table above, direct training refers to training SNN directly through the surrogate gradient method. In **Table 2** of our paper, work not marked with "TTFS-based" is rate-based method by default.
>
> [Wang et al. 2025] Ziqing Wang and Yuetong Fang and Jiahang Cao and Hongwei Ren and Renjing Xu, "Adaptive Calibration: A Unified Conversion Framework of Spiking Neural Network", AAAI 2025.
>
> [Yao et al. 2025] Man Yao, et al. "Scaling spike-driven transformer with efficient spike firing approximation training." IEEE Transactions on Pattern Analysis and Machine Intelligence (2025).
>
> [Zhou et al. 2024] Chenlin Zhou et al. "QKFormer: Hierarchical Spiking Transformer using Q-K Attention", NIPS 2024.
>
> [Yao et al. 2024] Man Yao, et al. "Spike-driven Transformer V2: Meta Spiking Neural Network Architecture Inspiring the Design of Next-generation Neuromorphic Chips.", ICLR 2024.
>
> ## 2. Discuss hardware implementation challenges and latency trade-offs of the proposed model could enhance the future work.
>
> ### Hardware Implementation Challenge
>
> We think that potential challenges in hardware implementation include:
>
> - Design special hardware that is compatible with the non-linear neurons proposed in our method. With different dynamics compared with traditional LIF models, the hardware needs a special design in order to fully demonstrate the energy efficiency of our proposed methods.
> - Deal with hardware-related accuracy loss. Although we have proved that our conversion method is theoretically lossless, there will inevitably exist hardware-specific loss, including noise in membrane potential and time bias of TTFS spikes. We have discussed the robustness of our proposed method in Section 5.3, and more analysis is needed for hardware implementation. Fine-tuning is probably needed if the hardware loss is too large.
>
> ### Latency Trade-offs
>
> In our proposed method, there are two separate stages for one layer, namely, the emitting stage and the receiving stage. As proposed in [Park et al. 2020], the two stages can be overlapped in order to reduce the latency, requiring a balance between the conversion loss and latency reduction. Since our work mainly focuses on a lossless conversion method of Transformer architecture, we leave the topic for future works.
>
> We will add more discussion in the revised version.
>
> ## 3. The TTFS could provide advantage in the aspect of low firing rates, hence there could supplement the firing rate comparison and analysis.
>
> Thank you for your inspiring question. Compared with previous work on TTFS, each non-linear neuron emits exactly one spike every forward pass. The firing rate is relatively lower than the rate-based method since multiple time steps is needed in rate-based SNN, leading to multiple spikes emitted in one neuron. However, the firing rate is higher than previous work on TTFS-based converted CNN, since we cannot simply ignore those negative values in the network with non-ReLU activation functions and attention mechanisms.
>
> The firing rate can be further cut down. For example, in activation function SiLU, we can regard $\mathrm{SiLU}(x) \approx 0$ for $x \le -16$ and simply emit no spike, since $\mathrm{SiLU}(-16) \approx -1.8 \times 10^{-6}$ is close to 0. In this way, we can still keep the information carried by values around zero while enhancing energy efficiency.
>
> [Park et al. 2020] S. Park, S. Kim, B. Na and S. Yoon, "T2FSNN: Deep Spiking Neural Networks with Time-to-First-Spike Coding," 2020 57th ACM/IEEE Design Automation Conference (DAC), San Francisco, CA, USA, 2020, pp. 1-6

---

### Official Review · Reviewer_4BzB · 2025-03-14

**Overall Recommendation:** 3

**Summary:**

The paper introduces a novel approach, TTFSFormer, for converting Transformer architectures into Spiking Neural Networks (SNNs) with Time-to-First-Spike (TTFS) coding. The method addresses the challenge of preserving high accuracy while significantly reducing energy consumption. The authors propose new neuron models and detailed conversion mechanisms to accommodate non-linear activations and complex components like the attention mechanism, which is a significant step in adapting SNNs to Transformer models. Experimental results on the ImageNet-1k dataset show that the method performs comparably to the original ANN architecture with minimal accuracy loss and lower energy costs.

**Claims And Evidence:**

no

**Essential References Not Discussed:**

no

**Ethical Review Flag:**

Flag this paper for an ethics review.

**Experimental Designs Or Analyses:**

he authors provide strong experimental results, showing that their method performs well on multiple Transformer architectures, including ViT and EVA, with minimal accuracy loss (below 0.1%).

**Methods And Evaluation Criteria:**

Novelty: This is the first work to focus on converting Transformer architectures to SNNs using TTFS encoding, which addresses both energy efficiency and accuracy preservation, two key challenges in SNNs.

**Other Comments Or Suggestions:**

Limited Novelty in SNN Design: The paper introduces some interesting neurons and neuron dynamics, but the fundamental idea of TTFS-based SNNs is not entirely new, as it has been explored in other contexts. More emphasis on how the Transformer-specific components are handled would add more value to the novelty claim

**Other Strengths And Weaknesses:**

Hardware Realism: The paper focuses primarily on the theoretical model and simulation results but lacks in-depth analysis of hardware implementation or real-world constraints, such as precision or latency in hardware. While the robustness of the model is mentioned, more detailed hardware evaluation would improve the practical impact of the work.
Scalability Concerns: While the method performs well on the tested architectures, the scalability of TTFSFormer to larger, more complex Transformers with more layers and parameters remains unclear. An exploration of how the method scales with model size and complexity would be valuable.
.

**Questions For Authors:**

no

**Relation To Broader Scientific Literature:**

While the paper compares TTFSFormer to other SNN methods, there is insufficient comparison with other Transformer-to-SNN methods that leverage different spiking encodings (e.g., rate coding or surrogate gradient methods). A more thorough comparative analysis with state-of-the-art techniques would enhance the paper’s contribution.

**Theoretical Claims:**

he paper makes an important theoretical contribution by introducing generalized nonlinear neurons for the conversion process, significantly expanding the applicability of TTFS coding to complex architectures like Transformers.

---

> ### Author Rebuttal · Authors · 2025-03-31
>
> Thank you for your thoughtful comments. We would like to address your concerns in the following.
>
> ### Hardware Implementation
>
> Thanks for your question. Since rate-based SNN is currently the most popular coding method, existing neuromorphic chips are specially designed for rate-based algorithms. With existing hardware circuits, one possible workaround is to store the mapping of non-linear functions in hardware storage, and simulate the change of potential. This is easy to implement since the non-linear function is shared in one layer. We hope that there will be neuromorphic chips suitable for TTFS-based SNN models, which will better demonstrate the energy efficiency of temporal coding methods. We will add more discussion in the revised manuscript.
>
> ### Scalability Concerns
>
> Different from other tasks such as NLP, the size of vision models typically ranges from 80M (such as ViT-B) to 1000M. In order to better demonstrate the scalability of our proposed method, we test our method on **EVA-G** (Giant) with 1.01B parameters in total, which is one of the largest popular pre-trained vision transformers. The experimentable result is shown in the table below.
>
> |  | EVA-G (ANN) | EVA-G (SNN) | EVA-G (SNN, with precision=1024) |
> | --- | --- | --- | --- |
> | Top-1 accuracy | 88.882\% | 88.898\% | 88.160\% |
> | Top-5 accuracy | 98.678\% | 98.684\% | 98.440\% |
>
> The result shows that our conversion method is still virtually lossless when converting a large vision model, indicating that our method has excellent scalability in vision task.
>
> ### Limited Novelty in SNN Design
>
>
> We agree that the idea of time-to-first-spike (TTFS) coding has been put forward and explored in previous works. Many SNN algorithms, either direct training or conversion, have created convolutional SNN successfully with much lower energy consumption and comparable performance to CNN. However, we would like to clarify that our work is the first to focus on the transformer model with TTFS coding.
>
> Our work mainly focuses on analyzing and solving the challenges encountered in introducing transformer architecture into TTFS-based SNN, which has not been addressed in previous works.
>
> - Challenge 1: Transformer architecture involves plenty of non-linear operations, such as non-ReLU activations, attention, and LayerNorm. However, previous TTFS neuron dynamics are only able to represent piecewise linear functions (Section 3.3).
> - Our Solution 1: We proposed a non-linear neuron model, which introduces non-linearity in SNN. We have theoretically proved that our neuron model has a strong representation ability (Section 4.2). With the proposed neuron model, we can construct Transformer-specific components (in Section 4.3).
> - Challenge 2: Previous TTFS neurons have a small representation range.
> - Our Solution 2: We expand the representation range of TTFS neurons depending on the actual distribution of the input and output values.
> - Challenge 3: Accuracy of converted models.
> - Our Solution 3: We have theoretically proved that our conversion method is lossless. Moreover, we conduct experiments on various vision transformer models (ViT, EVA) with different sizes (from 80M to 1B), showing that our method can achieve SOTA performance in vision tasks.
>
> ### Insufficient Comparison
>
> Thanks for your suggestion. We add the comparison with more recent works as follows.
>
> | Work | Method | ANN Model | Accuracy |
> | ---- | ------ | --------- | -------- |
> | Adaptive Calibration [Wang et al. 2025] | burst, conversion | ViT | 77.09\% |
> | E-Spikeformer [Yao et al. 2025] | rate, direct training | - | 86.2\% |
> | QKFormer [Zhou et al. 2024] | rate, direct training | - | 85.65\% |
> | Spike-driven V2 [Yao et al. 2024] | rate, direct training | - | 80.0\% |
> | **Ours** | TTFS, conversion | ViT-L/EVA02-L | 85.8\%/90.0\% |
>
> In **Table 2** of our paper and the table above, direct training refers to training SNN directly through the surrogate gradient method. In **Table 2** of our paper, work not marked with "TTFS-based" is rate-based method by default.
>
> [Wang et al. 2025] Ziqing Wang and Yuetong Fang and Jiahang Cao and Hongwei Ren and Renjing Xu, "Adaptive Calibration: A Unified Conversion Framework of Spiking Neural Network", AAAI 2025.
>
> [Yao et al. 2025] Man Yao, et al. "Scaling spike-driven transformer with efficient spike firing approximation training." IEEE Transactions on Pattern Analysis and Machine Intelligence (2025).
>
> [Zhou et al. 2024] Chenlin Zhou et al. "QKFormer: Hierarchical Spiking Transformer using Q-K Attention", NIPS 2024.
>
> [Yao et al. 2024] Man Yao, et al. "Spike-driven Transformer V2: Meta Spiking Neural Network Architecture Inspiring the Design of Next-generation Neuromorphic Chips.", ICLR 2024.

---

### Official Review · Reviewer_Cfcp · 2025-03-17

**Overall Recommendation:** 4

**Summary:**

This paper propose an ANN-SNN conversion method for spiking transformer based on time-to-first-spike (TTFS) method. The author first analyze the limitations of the previous TTFS method, and then propose a generalized TTFS neuron, which make it easier to relate Transformer to its SNN version. Experimental results on ViT and EVA models demonstrate the SOTA performance of the proposed method.

**Claims And Evidence:**

Yes, it is supported by rigorous theorem derivation and experimental analysis.

**Essential References Not Discussed:**

No.

**Experimental Designs Or Analyses:**

I have checked Section 5.1-5.3.

**Methods And Evaluation Criteria:**

Yes.

**Other Comments Or Suggestions:**

See weakness.

**Other Strengths And Weaknesses:**

Strengths:

1.The idea of using TTSF to implement spiking Transformer is impressive, which can reduce energy consumption in neuromorphic chips.

2.The proposed method achieve SOTA performance with ViT and EVA model.

3.The authors provide rigorous theoretic analysis.

Weakness:

1.The writing needs improvement. The article does not read very coherently, e.g., paragraphs 1-2 of the introduction.

2.The authors need to analyze the computational cost of the generalized TTFS neuron, especially compared to the LIF neuron.

3.This paper focused on improving classification performance of SNN, which in my view, is not the main strength of SNNs. The authors need to analyze other potential advantages. Actually, EVA can be used to other visual tasks. Whether the proposed method can be applied to other visual tasks?

**Questions For Authors:**

1.Please clarify how to compute the energy consumption in Table 2.

2.Could the proposed method be used to other model? Like CNN and ResNet.

3.Whether the proposed method can be generalized to the language task? I would like to see some discussion on this direction.

**Relation To Broader Scientific Literature:**

The contribution of this paper is related to spiking neural networks and low-power artificial intelligence.

**Theoretical Claims:**

I have checked the theorem and the proof.

---

> ### Author Rebuttal · Authors · 2025-03-31
>
> Thank you for your positive and thoughtful comments. We are encouraged that you find our idea impressive. We are glad you agree that our method achieves SOTA performance with low energy consumption. We would like to address your concerns and questions in the following.
>
> ### 1. Please clarify how to compute the energy consumption in Table 2.
>
> Thanks for your question. Our energy estimation follows the methodology proposed by [Nitin Rathi and Kaushik Roy,  2020]. For clarity, we analyze operations in a ViT block, categorizing them into **linear** (standard TTFS neuron dynamics, as in prior work) and **non-linear** (novel to our method, with hardware-dependent energy costs). Below is the list of operations (where d=384 or 768 or 1024 for ViT-S/B/L respectively):
>
>
> | Type | Notation of Parameters | Shapes | SNN Ops | ANN Ops |
> | :---: | :---: | :---: | :---: | :---: |
> | LayerNorm | (M, N): Matrix $X \in \mathbb{R}^{M \times N}$ | (197,d), (197,d) | $2 \cdot 197 \cdot (9d+1)$ | $2\cdot 197\cdot d$ |
> | MatMul | (M, K, N): Matrix $A \in \mathbb{R}^{M \times K},B \in \mathbb{R}^{K \times N}$ | (197,d,197), (197,197,d) | $197 \cdot [197 \cdot (3d + 2) + d \cdot (3\cdot 197+2)]$ | $2\cdot 197^{2}\cdot d$ |
> | SoftMax | (M, N): Matrix $X \in \mathbb{R}^{M \times N}$ | (197,197) | $6 \cdot 197^{2}$ | $197^{2}$ |
> | GeLU | (M, N): Matrix $X \in \mathbb{R}^{M \times N}$ | (197,4d) | $197 \cdot 4d$ | $197 \cdot 4d$ |
> | Linear | (M, K, N): Matrix $X \in \mathbb{R}^{M \times K}$ and weight $W \in \mathbb{R}^{K \times N}$ | (197,d,3d),(197,d,4d),(197,4d,d) | $11\cdot 197\cdot d^{2}$ | $11\cdot 197\cdot d^{2}$ |
>
> (The number of operations in each SNN layer is analyzed in section 4.3 and appendix C.)
>
> Now we can estimate the energy efficiency:
>
> $$
> \eta = \frac{\text{Energy of SNN}}{\text{Energy of ANN}}
> = \frac{(1206d+1578)E_{nl} + 11d^{2}E_{l}}{(11d^{2}+400d+197)E_\text{MAC}}
> \approx \frac{1206 \cdot E_{nl} + 11d \cdot E_{l}}{(11d + 400)E_\text{MAC}}
> $$
>
> where $E_\text{MAC}$ is the energy consumption of one multiply-add operation in ANN, $E_{l}, E_{nl}$ is the energy consumption of one linear and non-linear operation in SNN.
>
> We use the settings in [Horowitz, 2014], namely $E_{l} = 0.9 \text{pJ},E_\text{MAC}=4.6\text{pJ}$. We assume that $E_{nl} = kE_{l}$, where $k$ is a constant depending on the hardware implementation.
>
> Take ViT-L for example, by letting $d=1024$, we have $\eta = 0.189 + 0.020k$.
>
> In our paper, we assume that $k=1$, i.e. the non-linear neuron has the same cost with the linear neuron. We think that a reasonable estimation would be $E_{l} < E_{nl} < E_\text{MAC}$, for example $k=2$ or $k=3$. Although $k$ may vary according to the hardware implementation, we can conclude that the energy efficiency is between 20\%-30\%.
>
> ### 2. Could the proposed method be used to other models? Like CNN and ResNet.
>
> Thanks for your question. Our work generalizes prior TTFS-based methods from CNNs (consisting of linear, convolutional layers and ReLU) to Transformers. For CNN/ResNet architectures, our model reduces to the linear neuron dynamics described in **Corollary 4.4** and **4.6**, which is almost exactly what existing TTFS approaches (e.g. [Stanojevic et al. 2024]) are. While our method is compatible with CNN architectures, this would not introduce novel improvements for CNNs.
>
> ### 3. Whether the proposed method can be generalized to the language task? other visual tasks?
>
> Thanks for your inspiring discussion. While our experiments focus on vision tasks, the method is theoretically compatible with standard Transformer architectures in NLP. However, the generalization ability requires further experiments. For example, LLMs are typically larger than vision models, which poses challenges to the scalability of our method. We view this as a promising direction that desires future work. Besides, our method can be applied to regression-based vision tasks like object detection. We will add more discussion.
>
> ### 4. Computational cost of the generalized TTFS neuron
>
> The energy consumption of our non-linear neuron will be similar to or slightly higher than that of the LIF neuron. LIF neurons can be regarded as a special case of non-linear model where the potential change $\eta,\psi$ are exponential. When the potential change is not exponential, special hardware design is required, which is expected to have a similar energy cost with LIF neurons.
>
> [Nitin Rathi and Kaushik Roy, 2020] Nitin Rathi and Kaushik Roy. 2020. Diet-SNN: Direct Input Encoding with Leakage and Threshold Optimization in Deep Spiking Neural Networks.
>
> [Horowitz, 2014] M. Horowitz, "1.1 Computing's energy problem (and what we can do about it)," 2014 IEEE International Solid-State Circuits Conference Digest of Technical Papers (ISSCC), San Francisco, CA, USA, pp. 10-14
>
> [Stanojevic et al. 2024] Stanojevic, A., Woźniak, S., Bellec, G. et al. High-performance deep spiking neural networks with 0.3 spikes per neuron. Nat Commun 15, 6793 (2024).

---

### Decision · Program_Chairs · 2025-05-01

**Decision:**

Accept (poster)

**Comment:**

This paper proposes **TTFSFormer**, a method to convert Transformer models into **Spiking Neural Networks (SNNs)** using **Time-to-First-Spike (TTFS)** coding. The key innovation lies in enabling TTFS-based SNNs to handle the nonlinear operations characteristic of Transformers—e.g., GELU, SoftMax, LayerNorm—via newly designed spiking neuron dynamics with flexible kernel-based transformations. The method achieves minimal accuracy loss (typically <0.1%) while significantly reducing estimated energy consumption (~20%–30%), and is evaluated on models from ViT-S to EVA-G (>1B parameters).

## Strengths
- **Novelty and Scope**: This work demonstrates lossless TTFS conversion for Transformers, extending beyond CNN-focused TTFS methods. The neuron dynamics design is theoretically solid and expands the expressive power of TTFS neurons.
- **Theoretical Rigor**: The manuscript includes well-presented theorems (e.g., Theorems 4.1–4.3), proofs, and corollaries that are largely validated by Reviewer 5bT2. It clearly delineates how TTFSFormer handles each Transformer component.
- **Empirical Results**: Experiments on ImageNet-1k using ViT and EVA models demonstrate near-lossless accuracy and meaningful energy gains. Robustness to spike timing precision is also explored.
- **Rebuttal Quality**: The authors engaged comprehensively with all reviewer concerns, offering detailed clarifications, expanded hardware and latency discussions, additional comparisons to recent work, and plans for code release.

## Weaknesses
- **Writing Quality**: As highlighted by Reviewer Cfcp, the writing—especially in early sections—is at times incoherent and would benefit from thorough editing for clarity and flow.
- **Hardware Realism**: Several reviewers noted the limited discussion of hardware-specific implications (e.g., TTFS neuron latency, noise, or real-world spike timing). The authors responded with helpful elaboration, but empirical validation on actual neuromorphic hardware remains absent.
- **Baseline Comparisons**: While improved in the rebuttal, initial comparisons with recent rate-based or surrogate gradient SNN Transformers were sparse. This has been partly remedied with a supplemental table and additional discussion.

## Rebuttal Assessment
The rebuttal was **constructive and thorough**, addressing nearly all concerns with concrete analyses. Particularly commendable are the discussions around firing rate efficiency, scalability to larger models, and hardware implementation trade-offs. Several reviewers explicitly indicated that their concerns were fully resolved post-rebuttal.

This is a solid and well-motivated contribution to the field of energy-efficient neuromorphic computing and ANN-to-SNN conversion, particularly in expanding the applicability of TTFS coding to complex modern architectures like Transformers.